# Delayed Onset of Dural Arteriovenous Fistula Following Trauma

**William Richardson [1], Praveen Satarasinghe [2] and Michael T. Koltz [1,2,*]**

1   Seton Brain and Spine Institute, Round Rock, TX 78665, USA; will.richardson.1994@gmail.com
2   Department of Neurosurgery, Dell Medical School, University of Texas at Austin, Austin, TX 78712, USA; praveensatarasinghe@utexas.edu
*   Correspondence: mtkoltz@gmail.com; Tel.: +1-786-512-234-4816

**Abstract:** Dural Arteriovenous Fistulas (dAVF) are pathological shunts that are often idiopathic in presentation. However, it is reported that many patients presenting with dAVF have past medical histories notable for surgeries, hypercoagulation disorders, infections, and trauma. In trauma-linked dAVF, presentation generally occurs within 48 h post-incident. In the present case, the authors discuss the delayed onset of a Borden type II dAVF in a patient 12 hospital days post-trauma, as well as the course of treatment. This unique case provides a compelling demonstration for providers to be aware of the development of dAVF, even after the typical 48-hour post-trauma window. By being aware of the possibility of delayed dAVF presentation, delayed diagnosis or misdiagnosis can be avoided and emergent action can be taken.

**Keywords:** traumatic brain injury; arteriovenous; fistula; trauma; delayed; dAVF

## 1. Introduction

Dural arteriovenous fistulas (dAVF) are rare pathological shunts comprising only 10–15% of intracranial vascular malformations [1,2]. These malformations allow an exchange of blood between dural arteries and dural venous sinuses, meningeal veins, or cortical veins [3]. Depending on the location of occurrence, this exchange poses several risks, including ischemic stroke due to inadequate arterial blood supply to the tissues, as well as cerebral hematoma [4].

Relative risk of these lesions is evaluated by use of the Borden Classification System, which classifies dAVF into three types according to their venous drainage patterns and the presence or absence of cortical venous drainage [3,4]. Borden Type I lesions exhibit no cortical venous drainage and thus pose the least risk. Borden Type II and III lesions have cortical venous drainage which presents a greater risk of ischemic stroke or cerebral hemorrhage [5].

While the cause of dAVF remains uncertain and a multitude of presentations are idiopathic in nature, the health histories of many patients who present indicate there may be a link to infections, surgeries, hypercoagulation disorders, and trauma [3,5–7]. Typically, trauma-related dAVF occurs within 48 h of the incident, with no cases of delayed onset (>72 h post incident) reported in the English literature [1,7–9]. Here we present the case of a 49-year old male who developed an intracranial Borden Type II dAVF 12 days after initial trauma.

## 2. Case Presentation

The patient, a 49-year-old Hispanic male, originally presented through the emergency department after a worksite accident where he had fallen from a height of 10 feet onto concrete. His Glasgow Coma Score (GCS) on presentation was 15. A computerized tomography (CT) scan showed the patient had sustained multiple injuries, including a right skull fracture and a small, non-surgical subdural

hematoma in the right hemisphere (Figure 1). CT angiogram of the head and neck done on admission as part of our trauma centers admission protocol was negative for any associated vascular lesions.

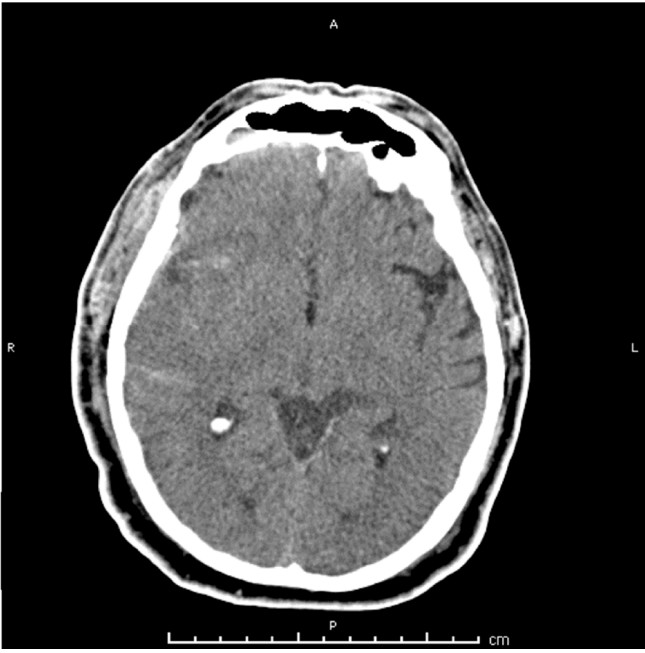

**Figure 1.** Computerized tomography (CT) head scan without contrast performed on admission, showing right hemispheric cortical subarachnoid hemorrhage and overlying rim acute subdural hematoma.

The patient developed aspiration pneumonia with respiratory distress requiring intubation on hospital day 2 but was extubated by hospital day 5 and remained GCS 15, however, the patient showed an acute deterioration in mental status on hospital day 12 with GCS 7 and left hemiplegia. CT scan was ordered, illustrating a large intracerebral hematoma. Emergent decompressive craniotomy and evacuation of the hematoma were performed (Figure 2).

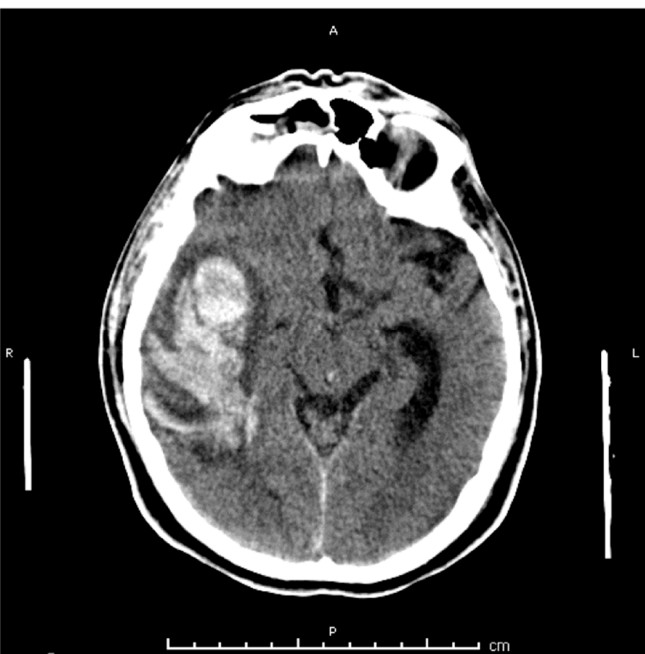

**Figure 2.** CT head scan without contrast performed on hospital day 12 after acute deterioration in mental status showing development of a large right temporal hematoma.

Large, dilated veins were observed intraoperatively, raising concern about the possible presence of a dAVF. To evaluate, the patient was taken for a cerebral angiogram 1-day post-surgery. A Borden Type II dAVF located between the middle meningeal artery (MMA) and cortical veins was identified (Figure 3). The lesion was successfully obliterated by injection of Onyx 18, 0.4 mL, as confirmed by performing post-treatment angiography (Figure 4).

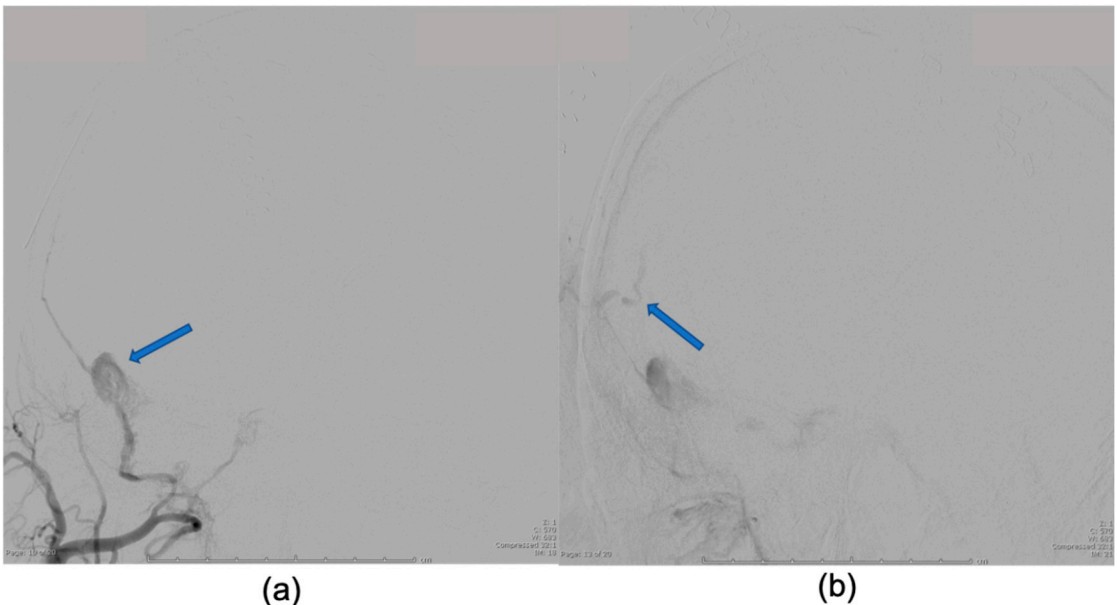

**Figure 3.** Digitally subtracted cerebral angiogram from external carotid artery, anteroposterior (AP) view done emergently on hospital day 12 showing middle meningeal artery (MMA) to cortical venous fistula as cause for delayed intracerebral hemorrhage. (**a**) Blue arrow pointing to MMA to cortical venous fistula site; (**b**) Blue arrow showing filling of cortical veins.

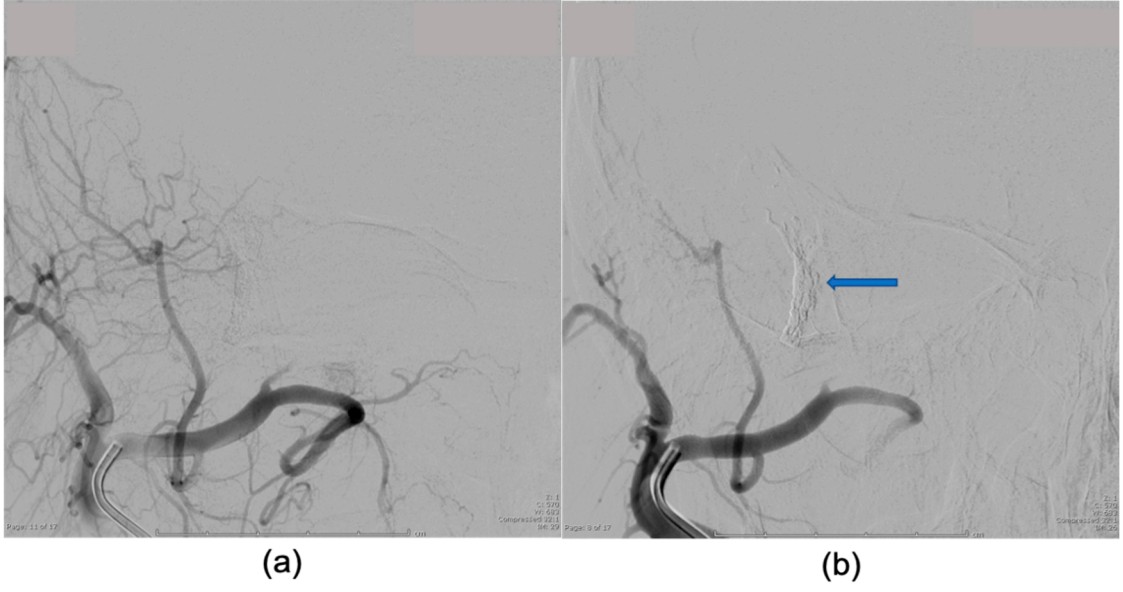

**Figure 4.** Digitally subtracted cerebral angiogram from external carotid artery, AP view performed emergently on hospital day 12 showing post treatment images after Onyx-18 embolization. (**a**) The previously seen middle meningeal artery (MMA) to cortical venous fistula is no longer filling with contrast and there is no intracranial cortical venous filling; (**b**) Blue arrow showing Onyx cast at fistula site.

　　　The patient recovered well from his emergent procedures and at the time of most recent follow up, post-operative day 35, had returned to GCS 15 with mild left sided neglect.

## 3. Discussion

　　　To the best of our knowledge, this is the first reported case of the delayed onset of a trauma-linked dAVF [1,6,7,9–11]. In addition, there are a limited number of reports on the subject in the literature and this report adds to the base of knowledge needed for further research into dAVF pathogenesis, presentation, and treatment.

　　　dAVF are rare lesions that allow for an exchange of blood between dural arteries, meningeal veins, cortical veins, or dural venous sinuses [1–3,9]. Incidents of trauma are known to be linked to dAVF pathogenesis [1]. In the literature, dAVF presentation typically occurs within 48–72 h following incident of trauma [7,8,12], however, the clinical and imaging features of dAVF are non-specific and diagnosis is often delayed or incorrect [13]. Clinical features include pulsatile tinnitus, visual disturbances, proptosis, seizures, and neurological deficits, and depend largely on the pattern of venous drainage and anatomical location [1,3,5,13]. For many of these clinical manifestations, MRI or CT are the preferred imaging tool for diagnosticians and will show proptosis/engorged ophthalmic vessels, intracranial hemorrhage, dilated intracranial vessels, venous outpouching, venous sinus thrombosis, etc. [3,7,14]. Until digitally subtracted angiography is performed, however, the presence of dAVF is typically not apparent [15]. In the current case, the delayed presentation of acute neurological deficit due to intracranial hemorrhage was non-specific as well, only being linked to the possibility of dAVF by the intraoperative observation of engorged veins. However, this was not confirmed until subsequent digitally subtracted angiography was performed. It appears that even in delayed cases, clinical and imaging features may be non-specific as well.

　　　Based on the pattern of venous drainage and presence or absence of cortical venous drainage (CVD) [3], dAVF are classified according to the Borden et al. system (Table 1) which helps explains the range of clinical manifestations [3,4]. The current patient was diagnosed with a type II Borden dAVF, which explains the acute decline in neurological function and large temporal hematoma.

**Table 1.** Borden Classification System for Dural Arteriovenous Fistulas (dAVF).

| Borden Classification | Venous Drainage | CVD [1] | Risk |
|---|---|---|---|
| Type I | Meningeal veins or dural venous sinus | No | Benign |
| Type II | Meningeal veins or dural venous sinus | Yes | Aggressive |
| Type III | Subarachnoid veins or venous sinus | Yes | Aggressive |

[1] CVD presents a risk factor for intracerebral hematoma.

　　　When considering the likelihood of the presence dAVF, the trauma team must be prudent in decision making and treatment to prevent dangerous and fatal conditions from developing. By being aware of the possibility of delayed dAVF development, clinicians can be more prepared. If symptoms indicating the possibility of dAVF arise, emergent action should be taken.

**Author Contributions:** Conception and design, M.T.K.; writing—original draft preparation, W.R.; writing—review and editing, P.S. and M.T.K.

**Conflicts of Interest:** The authors declare no conflict of interest.

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
