# Peer review of "Delayed Onset of Dural Arteriovenous Fistula Following Trauma"

_reports, doi:10.3390/reports2020014_

Round 1

Reviewer 1 Report

In this paper, the authors showed a case of right temporal intracerebral hemorrhage due to a ruptured dAVF clinically presented 12 days after a major head trauma. The suspect was intraoperatively raised and a post-op DSA confirmed the presence of a dAVF. The authors concluded supposing a pathophysiological mechanism based on a delayed onset of a dAVF following a head trauma. However, these data cannot support the authors' conclusions because it is not possible to exclude that the dAVF was already present at the time of the trauma.

Author Response

Reviewer 1,

Thank you for your thoughtful review of our case report. 

The patient had a CTA of the head and neck upon presentation to the emergency department as part of our trauma systems admission protocol and the final impression was  "A 3 mm rounded area of contrast accumulation adjacent to the anterior communicating artery most likely represents venous opacification. Small ACA aneurysm is not entirely excluded. No other aneurysm or vascular malformation is evident. Follow-up intracranial CTA may be of benefit."  

The lesion mentioned in the report, anterior communicating artery aneurysm, is an incidental finding and not in any way associated to the patients temporal lobe hemorrhage.  The admission CTA did not show any vascular lesions in the area of hemorrhage causing the patients neurologic decline.   

The hemorrhage pattern from the lesion mentioned in the report would be interhemispheric or basal cistern SAH.   

Reviewer 2 Report

This is a nicely reported case of dural arterovenous fistula occurred with 12 days of delay post-trauma.This case report deserves to be published. However, 1 or 2 statements on the final clinical outcome of the patient would improve greatly improve the manuscript. 

Author Response

Reviewer 2,

Thank you for your comments.  We have added information on the clinical course and outcome of the patient in lines 39-40, 42, 46-48, 50, 55-58, 70-71